# Diagnosis of Relapse of Colorectal Adenocarcinoma through CEA Fluctuation

**DOI:** 10.3390/reports7030060

**Published:** 2024-07-27

**Authors:** Zsolt Fekete, Patricia Ignat, Laura Gligor, Nicolae Todor, Alina-Simona Muntean, Alexandra Gherman, Dan Eniu

**Affiliations:** 1Oncology, “Iuliu Hațieganu” University of Medicine and Pharmacy, Cluj-Napoca 400012, Romania; suteu.patricia@umfcluj.ro (P.I.); gherman.alexandra@umfcluj.ro (A.G.); tudor.eniu@umfcluj.ro (D.E.); 2“Prof. Dr. I. Chiricuță” Oncology Institute, Cluj-Napoca 400012, Romania; todor@iocn.ro (N.T.); muntean.alina@yahoo.fr (A.-S.M.); 3County Emergency Hospital, Cluj-Napoca 400012, Romania; turcu_laura0803@yahoo.com

**Keywords:** CEA 1, colorectal cancer 2, follow-up 3, tumor markers 4, early intervention 5, early chemotherapy 6

## Abstract

Carcinoembryonic antigen(CEA) is a routine marker employed for follow-up of colorectal tumors. We aimed to determine whether a CEA increase within the normal range can be linked to a risk of recurrence. From the period of 2006–2013 we selected 78 consecutive patients with colorectal cancer, who underwent curative surgery with or without neo-/adjuvant chemo- or radiotherapy and had proper follow-up procedures. For analyzing CEA fluctuation, we used the smallest value of the CEA during follow-up as the reference. With the aid of a Chi-squared test, we have chosen the value of 1.1 ng/mL for significant CEA fluctuation. A total of 43.6% of patients had fluctuations in CEA of at least 1.1 ng/mL, with or without increases above 5 ng/mL. From these, in 79.4% of patients, the increases in CEA were explained either by recurrence (44.1%), adjuvant chemotherapy (20.6%) or benign pathology (14.7%). In 23% of the recurrences, a CEA increase of at least 1.1 ng/mL, but below 5 ng/mL, preceded the clinical relapse by a median of 8 months. Our conclusion is that an increase in CEA levels by at least 1.1 ng/mL within the normal range after curative treatment for colorectal cancer may serve as an early indicator of relapse or could be associated with other pathological conditions.

## 1. Introduction

According to GLOBOCAN, colorectal cancer ranks in third place in terms of incidence and second in terms of mortality worldwide [1]. In the situation of relapse, a multimodal aggressive approach, consisting of radical treatment of local or lymphatic relapse and/or oligometastatic sites, chemotherapy, molecular targeted therapy and immune therapy can rarely achieve a cure. If the intervention is early enough, it can offer a longer survival. Therefore, early diagnosis and timely therapeutic intervention for relapse seems to be crucial. Circulating tumor cells, novel protein tumor markers or serum genetic markers likely represent the future of more sensitive follow-up.

CEA (carcinoembryonic antigen) and CA 19-9, although not sensitive enough for screening and early diagnosis, are the two standard tumor markers used (1) at diagnostic workup (for prognosis) and (2) in post-treatment follow-up (for detection of a relapse). Among the two, CEA is the most utilized and its periodical use as an indicator of relapse has been well established in several prospective studies, even rendering routine periodic CT scans unnecessary based on some data [2,3]. For follow-up, after curative multimodal treatment of ≥T2 or N+ tumors, measurement is mandatory, usually every 2–3 months in the first 2–3 years and then every six months, until reaching 5 years [4]. Even if the initial CEA value falls in the normal range, it is still worthy to follow-up its values because it can still increase in the event of a relapse, as shown in an analysis of the Dutch TME trial data [5].

The normal values for CEA are considered <5 ng/mL (µg/L) by most guidelines and laboratories. Some guides go as low as <2.5 ng/mL or <3.4 ng/mL in non-smokers and <4.3 ng/mL in smokers, but usually only heavy smoking can increase CEA values above 3.4 ng/mL [6,7,8].

In a Cochrane review [9], which included 52 studies, the authors analyzed the sensitivity and specificity of CEA testing to detect postoperative relapses in the follow-up period. In the seven studies reporting the impact of applying a threshold of 2.5 ng/mL, the sensitivity and specificity were 82% and 80%, respectively. In the 23 studies reporting a threshold of 5 ng/mL, the sensitivity was 71% and the specificity increased to 88%. In the remaining seven studies reporting the impact of applying a threshold of 10 ng/mL, the pooled sensitivity was 68% and the specificity increased even more, to 97%. We can observe that the sensitivity was higher at a lower cut-off, of 2.5 ng/mL, while, naturally, the specificity increased to almost 100% at 10 ng/mL. We can also observe that even at a cut-off of 2.5 ng/mL, 18% of the relapses went undetected. In none of these studies was a postoperative (new baseline) carcinoembryonic antigen (CEA) value taken into account.

Wang et al. [10], in a recent meta-analysis, found a lower sensitivity for CEA, at a cut-off of 5 ng/mL of only 59%. The pooled sensitivity was high, at 89%. At such a low sensitivity, there is room for improving the use of CEA in the detection of a relapse.

The aim of this study was to analyze the fluctuation in CEA and to determine whether a certain increase in the conventional normal range of values can predict clinical recurrence (local recurrence, regional recurrence—i.e., lymph node recurrence—or metastasis).

The secondary objective of this study was to assess whether the post-surgery CEA value significantly impacts disease-free survival.

## 2. Materials and Methods

Our retrospective observational study was performed at a tertiary academic cancer center and included consecutive patients with rectal and non-rectal colon cancer diagnosed between January 2006 and December 2013 who underwent curative surgical treatment with or without neoadjuvant/adjuvant chemo- or radiotherapy. The reason for selecting this period was that during that time, all CEA measurements were conducted at the same laboratory using the same measurement method.

The inclusion criteria were as follows:Primary confirmed colorectal adenocarcinoma;Clinical stage II and III;Curative surgical treatment;Negative resection margins (R0);Post-surgery follow-up of the CEA values;

The exclusion criteria were as follows:Multiple synchronous colorectal tumors;Clinical stage IV;Other cancers (several other tumors express and secrete CEA).

The database used for this study was selected by browsing through 2620 files pertaining to patients treated with colorectal cancer from our institution to identify subjects who fulfill the inclusion/exclusion criteria. As a result, only 78 patients were selected from this period.

We collected all available CEA values measured before any treatment (the highest value was then considered) and during the postoperative follow-up. CEA values closer than one week after colonoscopy were eliminated. The first post-operative measurement of CEA was performed within 1 to 3 months after surgery. CEA was measured in the same laboratory using the same method by electrochemiluminescence immunoassay (ECLIA). CEA is considered normal in our laboratory at values < 3.4 ng/mL in non-smokers and <4.3 ng/mL in smokers. For the purpose of this study, however, we used the convention of <5 ng/mL for CEA values that are considered “normal”. As a reference for the fluctuations, we used the smallest value of CEA observed during follow-up. CEA values were not censored in the relationship with postoperative chemotherapy.

All data were collected in a FileMaker database and analysis was performed through Excel Microsoft Office (Version 2007).

The 4-year overall survival and disease-free survival (DFS) were determined through the Kaplan–Meier method. Survival differences were evaluated through the log-rank test.

The confidence intervals were estimated at the 95% confidence level. Statistical significance was defined by a value of *p* ≤ 0.05.

## 3. Results

### 3.1. General Characteristics of Patients

The results of this study are reported following the guidance of the Equator Network [11], specifically the recommendations of Rubino et al. [12]. From the 78 patients included in the study, 49 were men (69%). Patients’ ages ranged between 25 and 79 years and the median age was 59 years. The median follow-up was 42.1 months (range 12.4–93.1). Patient characteristics are shown in Table 1. The body mass index (BMI) at diagnosis ranged from 17.3 kg/m^2^ to 43 kg/m^2^, with a median of 26.2 kg/m^2^, with 39.7% of patients being overweight or obese. Out of the 60 patients with rectosigmoid cancer, 39 underwent anterior resection and 21 abdominoperineal resection. Out of eighteennon-rectal colon cancer patients, 10 underwent segmental resections, three left hemicolectomy, three transverse colectomy and two right colectomy. During surgery, a minimum 5 cm margin of normal bowel tissue was obtained on both sides of the tumor to reduce the risk of anastomotic recurrence. In cases of rectal cancer, the distal macroscopic margin was at least 1 cm. All resections were microscopically confirmed as R0.

### 3.2. Relapses

From the entire group of patients, 22 (28.2%) presented recurrences, of which most had metastasis as the only type of relapse (12 patients, 54.5%), 7 patients (31.8%) had both metastasis and local recurrence, and 3 patients (13.6%) had only local recurrence.

To analyze the association between the increase in CEA and tumor recurrence, we searched with a Chi-squared test to identify an “alarm value” for CEA, defined as the first increase in at least a certain value registered during follow-up, compared to the smallest previous value. We defined clinical recurrence as either local, regional or distant relapse, documented by imaging studies. For patients with CEA increase in more than this ”alarm value” without documented recurrence, we searched the patients’ records for possible benign causes of the fluctuation.

After analyzing all possible causes of an increase in CEA (relapse, chemotherapy and benign diseases), we chose the value of 1.1 ng/mL as the best “alarm value” because it was associated with the best p value (*p* = 0.0000139). The Chi-squared test can be observed in Figure 1.

Overall, 34/78, 43.6%,patients had fluctuations in CEA of at least 1.1 ng/mL. Of these 34 patients,27 (79.4%) increases in CEA were explained by recurrence, adjuvant chemotherapy or benign pathology. In 5 of 22 relapsed patients (22.7%), we observed an increase in the CEA values of at least 1.1 ng/mL, during follow up, before the clinical recurrence, with values positioned in the normal range interval. (Table 2). Notably, the CEA increase of at least 1.1 ng/mL preceded the clinical relapse by a median of 8 months (range 3–22 months).

In Figure 2, we show the fluctuation in CEA with an increase of at least 1.1 ng/mL but below 5 ng/mL in patients with clinical recurrence.

Seven patients out of the twenty-two(31.8%) presented relapse but had no increase in CEA either above 5 ng/mL nor with the defined alarm value of ≥1.1 ng/mL. The median of the peak CEA values was 1.63 ng/mL (range 1.39–3.38). The remaining subjects with clinical recurrence (10/22, 45.5%) presented an abrupt increase in CEA above 5 ng/mL with no preceding alarming increase of more than 1.1 ng/mL. Altogether, 15/34 (44.1%) CEA increases ofat least 1.1 ng/mL (either gradually or abruptly increasing above 5 ng/mL) were related to clinical recurrence.

### 3.3. Other Causes for CEA Fluctuations

Out of the 78 patients, 56 subjects (71.7%) had no recurrence during follow-up. Still, 19 patients out of these 56 (33.9%) presented an increase in CEA that exceeded 1.1 ng/mL. In seven patients (36.8%), the only evident cause for this fluctuation was adjuvant chemotherapy (Table 3 and Figure 3.)

The transient growth of CEA values occurred either during chemotherapy (five patients), or in a minority (two patients), the increase was detected one month or maximum two months post chemotherapy. In six out of seven cases, the values returned to “normal” (<5 ng/mL). In 12/34 of the patients with a fluctuation of at least 1.1 ng/mL, this was neither due to relapse nor to adjuvant chemotherapy (35.3% of patients with fluctuation); a possible association between the growth of the marker and a benign pathology was found in five patients, who were fully investigated for all possible benign pathology such as *colorectal adenomas* and *liver disease,* to name just the most frequent possible causes (Table 4, Figure 4).

A total of 44/78 (56.4%) patients had no increase in CEA values of at least 1.1 ng/mL and no relapse.

### 3.4. Summary of CEA Fluctuations

Among the 78 patients, 34 individuals (equivalent to 43.6%) exhibited fluctuations in carcinoembryonic antigen (CEA) levels exceeding 1.1 ng/mL. Among these 34 patients, 27 (or 79.4%) had CEA level increases that could be attributed to factors such as recurrence, adjuvant chemotherapy, or benign pathology. The odds ratio of a relapse in the presence of a CEA increase of at least 1.1 ng/mL was 4.17 (95% CI 1.45–11.97, *p* = 0.0079) (Table 5).

The estimated overall survival at 48 months (4 years) and disease-free survival rate, for the same period is shown in Figure 5.

### 3.5. Post-Surgery CEA Value and Disease-Free Survival

To analyze the influence of post-surgery CEA on DFS, we chose acut-off value of 2.5 ng/mL with the aid of an ROC curve and Chi−squared test (using Yate’s correction). The cut- off value was chosen by the minimum distance of the ROC curve to point (0.1). The odds ratio was calculated with the Chi-squared test. The disease-free survival was significantly influenced by the post-surgery value of CEA. Patients with CEA values < 2.5 ng/mL had a disease-free survival (DFS) at 4 years of 89%, while patients with values > 2.5 ng/mL had a DFS rate of only 55% (Figure 6).

## 4. Discussion

The disease-free survival was significantly influenced by the post-surgery value of CEA, being almost double in patients with values < 2.5 ng/mL compared to patients who had values > 2.5 ng/mL. Several studies have demonstrated that the pre-treatment value of this marker affects survival (values > 4–6 ng/ml) [13,14]. However, only scarce data are available about postoperative levels. In this study, we demonstrated a correlation between disease-free survival and the “post-curative” value of CEA. This result is similar to that published in the post hoc analysis of the MOSAIC and PETACC-8 trials. In this analysis, the 3-year DFS rate was 75%, 65% and 45% in a group of patients with CEA levels of 0–1.30 ng/mL (n = 630), 1.30–5 ng/mL (n = 613) and >5 ng/mL (n = 49). [15]

A new concept in rectal cancer prognostication is the CEA to MRI tumor volume ratio (CEA/Vol_MRI_). In the study reported by Zeng et al. [16], higher CEA/Vol_MRI_ was associated with worse DFS and OS. CEA/Vol_MRI_was superior to CEA, CEA and diameter on pathology ratio, and CEA and diameter on MRI ration in predicting DFS and OS.

Metastases as the only form of relapse represented the majority (54%) among patients with treatment failure, followed by metastasis along with local recurrence; meanwhile, the fewest subjects had only local recurrence. These data are consistent with the literature, with metastasis alone being the leading cause of treatment failure [17].

Five patients (22.7%) with clinical recurrence presented an earlier increase in CEA, although its value remained in the standard normal range of values (<5 ng/mL for smokers). The ability to suspect a relapse at lower values of CEA is important, since CEA increases gradually, as shown in the Dutch TME clinical trial: when relapse was diagnosed during follow-up, CEA values were normal at the first measurement in 81% of patients at a threshold of 5 ng/mL and in 66% at a threshold of 2.5 ng/mL.

Seven patients, 31.8% of the clinically relapsed patients, did not present neither fluctuations in CEA > 1.1 ng/mL nor increases above 5 ng/mL, underlining the need forcombined CEA-CT follow-up. These relapsed patients lack CEA secretion above a certain arbitrarily defined level. CEA cellular expression and secretion is around 50–70% [18], depending on primary or relapsed cancer, and thus cannot be used as an universally sensitive tumor marker. In our study, “CEA-silent” relapsed patients had CEA values of less than 3.4 ng/mL.

Nineteen patients had increases in CEA of at least 1.1 ng/mL without presenting relapse. These elevations were transitory, even though in some cases the values exceeded 5 ng/mLby far. Only 5/21 patients underwent general investigations to determine non-oncological causes of CEA growth. The causes of CEA increase in these cases were colon adenoma, cholecystitis, ulcerative colitis and antral gastritis. These were all benign diseases which were worth addressing with treatment; in other words, the alarm value of 1.1 ng/mL can be useful in these pathologies as well.

A study of the literature shows that several types of benign pathologies, including gastric, liver and lung diseases, and premalignant lesions can result in an increase in CEA over normal values in the absence of a malignancy [19,20,21,22]. The values in these cases are rarely over 10 ng/mL, although not exceptional [19]; in our study, there were only two patients with values over 10 ng/mL. In our findings, in almost all cases, there was a subsequent return to the normal range of values after transient increases.

Hypothyroidism can be a cause for abnormal CEA and TSH and fT4 should be measured if there is an otherwise unexplained increase in CEA [23]. Age and even blood groups influence CEA but its values are rarely over 3.4–5 ng/mL without an underlying pathology [24]. There are proponents of an adjusted CEA value based on age, BMI, WBC count, Hb, fasting glucose, AST, creatinine, triglyceride and HbA1c levels [25].

Another possible cause to take into consideration when examining CEA elevations is adjuvant chemotherapy (7/19 patients of fluctuations with 1.1 ng/mL and no relapse in our study). A hypothesis that would explain this phenomenon is the release of CEA from the apoptotic cells during chemotherapy, similar in patients receiving palliative chemotherapy. Another more likely explanation for the CEA increase could be the gastrointestinal toxicity or liver toxicity of chemotherapy. (Physiologically, the liver clears CEA efficiently from the serum.) Mitchell et al. also observed a transient increase in CEA during adjuvant chemotherapy with influence on the relapse risk [26].

The study has several limitations, including its retrospective design, the absence of investigation into fluctuations in all patients with a fluctuation of at least 1.1 ng/mL, and a lower patient count due to high exclusion rates resulting from improper follow-up procedures.

We have initiated a prospective clinical study of more detailed investigations prompted by a fluctuation in CEA of at least 1.1 ng/mL.

Lateral lymph node metastasis (LLNM) is an important prognostic feature in locally advanced rectal cancer [27]. In our study, none of the sixty rectal cancer patients exhibited LLNM on MRI, making our study homogeneous from this perspective.

Surgery followed by adjuvant chemotherapy (ACT) is the standard of care for patients with non-metastatic stage III and high-risk stage II colon cancer. In the case of rectal cancer, chemotherapy is preferentially used before tumor resection. Both adjuvant and neoadjuvant chemotherapy target microscopically disseminated disease, and in some cases, they successfully eliminate residual tissue. However, there are instances where tumor deposits persist, and some surviving cells exhibit stemness characteristics. A comprehensive review by Martínez-Pérez [28] highlights targeted therapies aimed at eradicating cancer stem-like cells (CSCs) in colorectal cancer. These therapies focus on CSC biomarkers, stemness-associated pathways, the tumor microenvironment, and immunoevasion, with the goal of improving cancer treatment outcomes and overcoming CSC-mediated therapy resistance.

Among other potentially useful biomarkers in colorectal cancer, we can take into account the tumor marker CA 19-9, circulating tumor cells (CTCs), and circulating tumor DNA (ctDNA).

Our research group has previously demonstrated that CA 19-9 levels can rise independently from CEA in about 15% of cases, and we have specifically observed this increase in grade 3 colorectal adenocarcinoma [29]. In the current study, CA 19-9 was not methodically measured.

In recent years, CTC detection technology has seen significant progress, utilizing methods like SE-iFISH and image flow cytometry. As a result, the CTC detection rate has improved. CTC positivity shows potential as a marker for detecting residual disease and guiding decisions on adjuvant chemotherapy in CRC patients.

To enhance CTCs’ ability to predict CRC recurrence, researchers are exploring various strategies. These include not only considering CTC subtypes but also combining CTC-related parameters with clinicopathological factors (e.g., tumor severity, lymph node metastasis, vascular invasion) and tumor markers (e.g., CEA, CA72-4, CA 19-9) to establish a multimarker model for predicting CRC recurrence. The application of this new strategy in clinical practice warrants further exploration.

In addition to CTCs, ctDNA has gained much attention as another ‘liquid biopsy’ marker. Unlike circulating tumor cells (CTCs), which primarily represent metastases-initiating cells, ctDNA offers additional insights into mutations and genomic alterations across the entire tumor genome. While CTCs are whole tumor cells captured in the bloodstream, ctDNA consists of fragments of DNA shed by apoptotic tumor-derived cells. ctDNA may serve as a precise tool for monitoring cancer relapse during CRC follow-up [30].

## 5. Conclusions

An increase in CEA of at least 1.1 ng/mL, detected in post-surgery follow up, even though in the normal range of values, should raise the hypothesis of a relapse and prompt close monitoring of patients, since it can predict clinical recurrence several months in advance. Other causes, such as adjuvant chemotherapy, gastric pathology and adenomas, are important factors to be taken into consideration in post-surgery follow up regarding CEA dynamics, causing transient increases in values even above 5 ng/mL. Disease-free survival is significantly influenced by the postoperative value of CEA with a cut-off value of 2.5 ng/mL.

## Figures and Tables

**Figure 1 reports-07-00060-f001:**
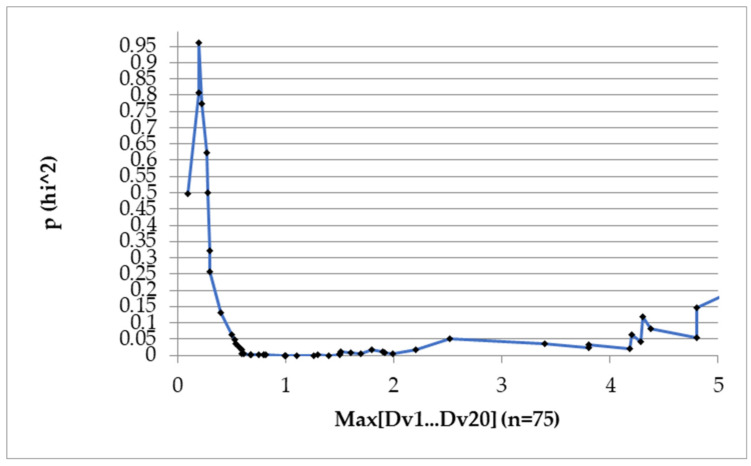
Chi-squared test to identify an alarm value for the increase in CEA (the difference between the smallest value and the highest value which would be associated with an event, i.e., relapse, chemotherapy, benign causes of increase in CEA. x: *p* values, y: CEA fluctuations in ng/mL.

**Figure 2 reports-07-00060-f002:**
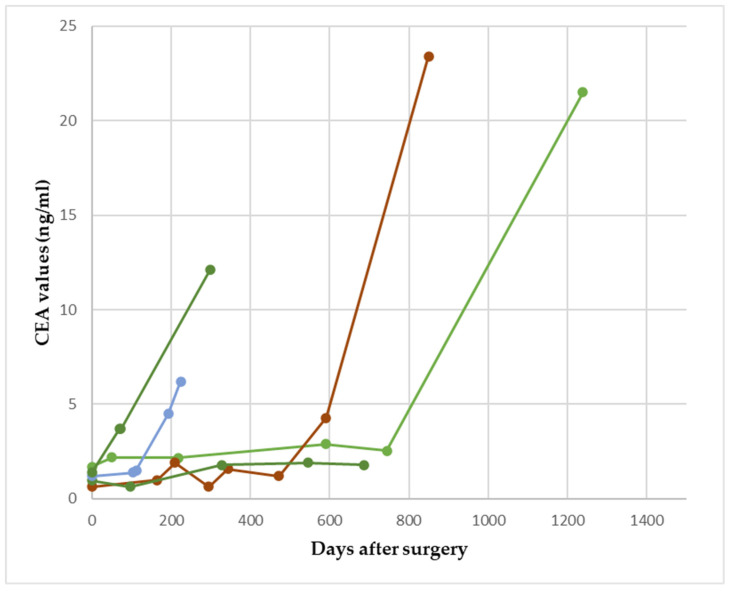
Increases in CEA below 5 ng/mL but at least 1.1 ng/mL, which predicted relapse in 5/22 patients. The colored lines represent different patients.

**Figure 3 reports-07-00060-f003:**
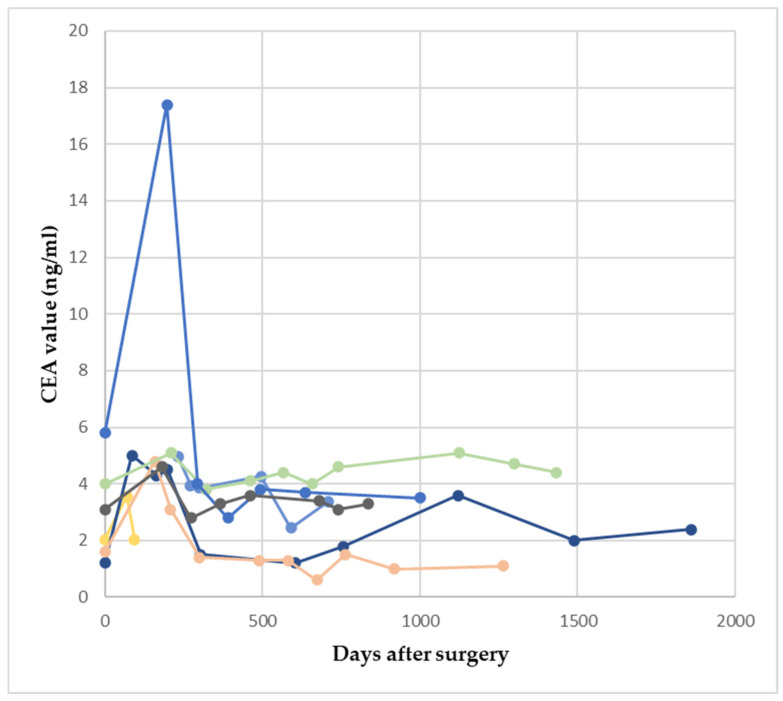
CEA fluctuation of at least 1.1 ng/mL under adjuvant chemotherapy. Colored lines represent patients.

**Figure 4 reports-07-00060-f004:**
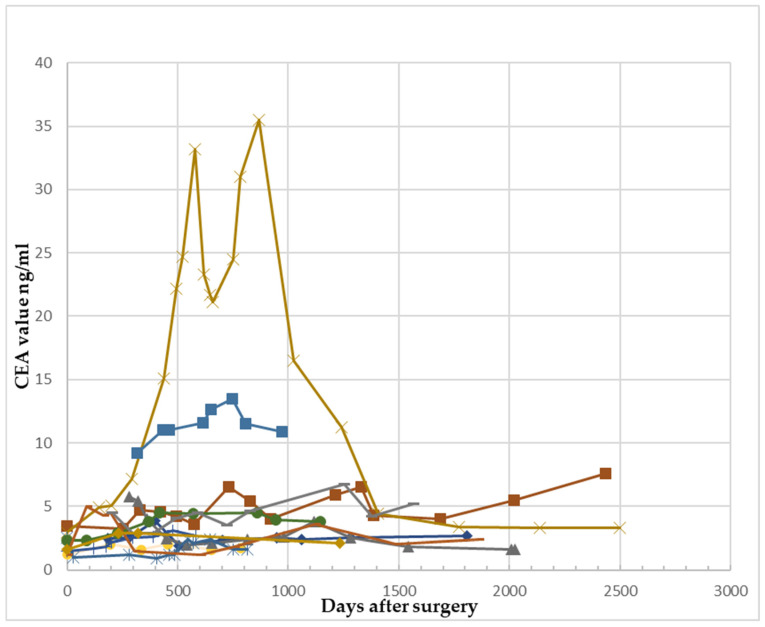
CEA fluctuation of at least 1.1 ng/mL for patients with no relapse and no adjuvant chemotherapy. Different colored lines designate different patients.

**Figure 5 reports-07-00060-f005:**
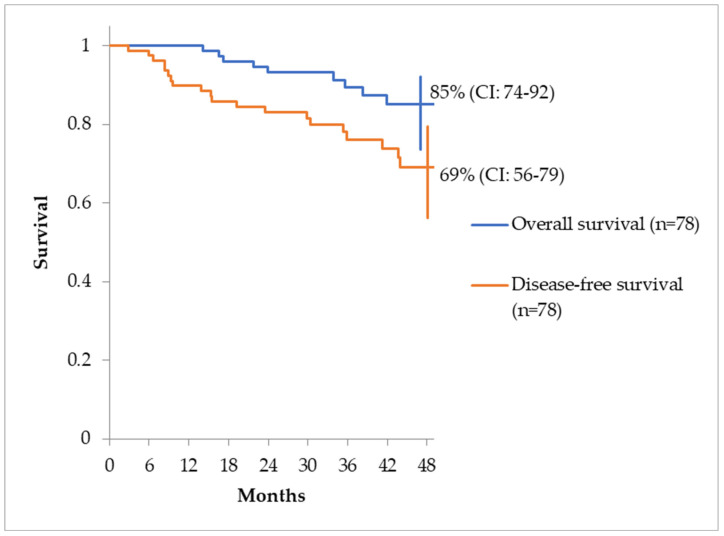
OS and DFS at 4 years.

**Figure 6 reports-07-00060-f006:**
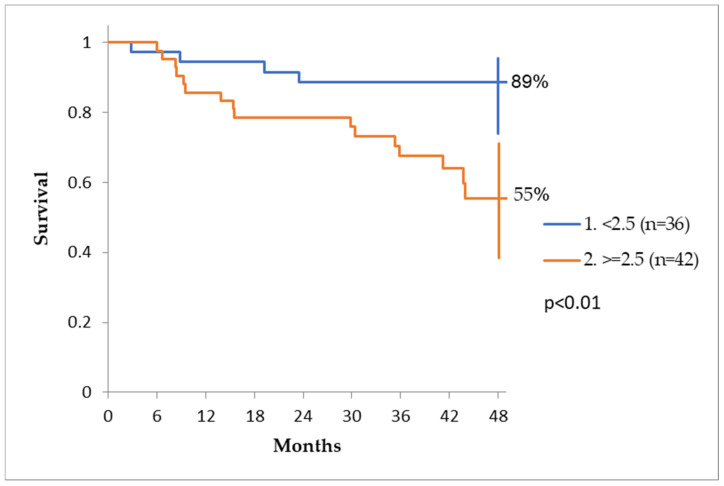
The post-surgery value of CEA marker and the DFS.

**Table 1 reports-07-00060-t001:** Patient characteristics.

Variable	Number	(%)
Sex		
Male	49	69
Female	29	31
Subsite Rectosigmoid Non-rectal colon Tumor stage	60 18	76.9 23.1
IIA IIB IIIA IIIB IIIC Neoadjuvant/Adjuvant chemotherapy Neoadjuvant/Adjuvant radiotherapy	26 2 2 41 7 51 50	33.2 2.6 2.6 52.6 9 65.4 64.1

**Table 2 reports-07-00060-t002:** Patients with increase in CEA of at least 1.1 ng/mL, but not reaching 5 ng/mL, before clinical recurrence.

Patient Number	Date of Clinical Recurrence on CT/MRI/PET-CT/US/Endoscopy	Baseline Value after Surgery	Alarm Value	Date of Alarm Value	Difference of at Least 1.1 ng/mL
6.	21.09.2009	1.70	2.89	13.12.2007	1.19
8.	01.11.2008	0.63	1.91	29.01.2007	1.28
24.	11.05.2009	0.64	1.90	03.12.2008	1.26
37.	02.02.2009	1.40	4.50	08.12.2008	3.10
78.	01.02.2015	1.40	3.70	05.06.2014	2.30

**Table 3 reports-07-00060-t003:** Increase in CEAofat least 1.1 ng/mL associated with adjuvant chemotherapy.

Patient Number	Minimum Value	Maximum Value	Date of First Increase with >1.1 ng/mL	Date of Start of Chemotherapy	Last Chemotherapy Cycle
13.	2.45	2.52	07.12.2006	14.08.2006	30.10.2006
28.	2.04	1.51	20.11.2007	11.10.2007	05.03.2008
43.	1.20	3.80	08.07.2009	26.05.2009	28.10.2009
54.	4.00	1.10	29.09.2011	30.03.2011	24.08.2011
63.	3.10	1.50	21.11.2011	25.07.2011	21.11.2011
50.	1.60	3.20	20.06.2011	07.02.2011	07.06.2011
55.	5.80	11.60	19.08.2011	04.03.2011	29.07.2011

**Table 4 reports-07-00060-t004:** CEA increase of at least 1.1 ng/mL for patients with no relapse and no adjuvant chemotherapy (probably benign causes and non-relevant fluctuations).

Patient Number	Possible Cause for CEA Fluctuation	Min. Value	Max. Value	Difference of at Least 1.1 ng/dL
9.	Not investigated for all benign causes	2.49	3.83	1.34
4.	Not investigated -//-	3.22	4.69	1.47
16.	Not investigated -//-	1.60	5.78	4.18
19.	Ulcerative colitis	2.97	35.50	32.5
18.	Colic adenoma	1.00	2.40	1.40
53.	Cholecystitis	2.30	4.50	2.20
75.	Not investigated -//-	1.50	3.10	1.60
44.	Not investigated -//-	1.20	3.60	2.40
45.	Not investigated -//-	2.90	4.50	1.60
59.	Not investigated -//-	1.60	2.90	1.30
61.	Gastritis	9.20	13.5	4.30
70.	Colic adenoma	2.70	8.40	5.70

**Table 5 reports-07-00060-t005:** CEA fluctuation and clinical recurrence.

Variable	Clinical Recurrence	No Clinical Recurrence	Total
CEA fluctuation	15	19	34
No CEA fluctuation	7	37	44
Total	22	56	78

OR 4.17 (95% CI 1.45–11.97, *p* = 0.0079).

## Data Availability

The data presented in this study are available on request from the corresponding author. The data are not publicly available due to patient privacy.

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
