# Peer review of "Diagnosis of Relapse of Colorectal Adenocarcinoma through CEA Fluctuation"

_reports, 2024, doi:10.3390/reports7030060_

Round 1
Reviewer 1 Report
Comments and Suggestions for Authors
In this manuscript, the authors investigated CEA fluctuation of colorectal cancer patients after surgery, and concluded that over 1.1 ng/ml of CEA levels is an indicator for relapse. The manuscript is well described, but some issues also need to be addressed for improving the quality of manuscript.
1. In the Introduction, the authors should summary the advances in CEA levels as indicators for colorectal cancer relapse;
2. In the Materials and Methods, determination of CEA should be described in details;
3. Add a “,” after In Figure 2 in Line 131;
4. Add unit for 1.1 in Line 225;
5. There are many paragraphs only contain one sentence, it should be combined;
6. For reference, some journal name is abbreviated, some are full journal name, it should be consistent.
Comments on the Quality of English LanguageIt is fine.
Author Response
Comments 1:
In this manuscript, the authors investigated CEA fluctuation of colorectal cancer patients after surgery, and concluded that over 1.1 ng/ml of CEA levels is an indicator for relapse. The manuscript is well described, but some issues also need to be addressed for improving the quality of manuscript.
- In the Introduction, the authors should summary the advances in CEA levels as indicators for colorectal cancer relapse;
- In the Materials and Methods, determination of CEA should be described in details;
- Add a “,” after In Figure 2 in Line 131;
- Add unit for 1.1 in Line 225;
- There are many paragraphs only contain one sentence, it should be combined;
- For reference, some journal name is abbreviated, some are full journal name, it should be consistent.
Response 1:
We appreciate your time and dedication to thoroughly analyze our article. We have tried as much as we could to address all your concerns and made the necessary adjustments as well. Please find our responses below. We have edited our manuscript accordingly.
- We have added more information, two systematic reviews in the Introduction section and one more individual study in the Discussion section. Changes are marked with red.
- We have added more details.
- Added a coma in line 131.
- Added the unit ng/ml.
- Thank you for your suggestion, we have combined several short paragraphs now.
- We reviewed and abbreviated the journal names and respected standard abbreviations.
Sincerely,
Dr Fekete

Reviewer 2 Report
Comments and Suggestions for Authors
This is an interesting study concerning the diagnosis of CRC recurrence through CEA fluctuation.
In have the following comments:
- Why did the authors select this specific time period? (which should also be specified in the abstract)
- The correct checklist for the specific type of article must be added
please see equator network
- The limitations of the study must be added
It would be useful to also highlight the role of other biomarkers or factors that could be associated with CEA. Please report the following articles
- Prognostic value of lateral lymph node metastasis in pretreatment MRI for rectal cancer in patients undergoing neoadjuvant chemoradiation followed by surgical resection without lateral lymph node dissection: A systemic review and meta-analysis. Eur J Radiol. 2024 Jul 2;178:111601. doi: 10.1016/j.ejrad.2024.111601
- Targeted Treatment against Cancer Stem Cells in Colorectal Cancer. Int J Mol Sci. 2024 Jun 5;25(11):6220. doi: 10.3390/ijms25116220
Comments on the Quality of English Language
Moderate
Author Response
This is an interesting study concerning the diagnosis of CRC recurrence through CEA fluctuation.
In have the following comments:
- Why did the authors select this specific time period? (which should also be specified in the abstract)
- The correct checklist for the specific type of article must be added, please see equator network
- The limitations of the study must be added
- It would be useful to also highlight the role of other biomarkers or factors that could be associated with CEA.
- Please report the following articles
- Prognostic value of lateral lymph node metastasis in pretreatment MRI for rectal cancer in patients undergoing neoadjuvant chemoradiation followed by surgical resection without lateral lymph node dissection: A systemic review and meta-analysis. Eur J Radiol. 2024 Jul 2;178:111601. doi: 10.1016/j.ejrad.2024.111601
- Targeted Treatment against Cancer Stem Cells in Colorectal Cancer. Int J Mol Sci. 2024 Jun 5;25(11):6220. doi: 10.3390/ijms25116220
Dear Reviewer/ Dear Colleague,
Your thorough and constructive feedback has been received with much enthusiasm. Our detailed responses are provided below, addressing each point directly.
- In the selected period all CEA measurements were performed at the same laboratory (Synevo), who had a contract for the entire period with our academical center. We have added now this information (and specified the period in the abstract as well).
- We have made several small adjustments to our article after consulting the Equator Network’s guidelines. Specifically, we have used Rubino M. e at al., Guidelines for reporting case series of tumours of the colon and rectum. Changes are marked with red.
- We have added further limitations. Changes are marked with red.
- We have added the value of CA-19-9, CTCs and ctDNA.
- We have included the studies in the Discussion. Thank you for your suggestions!
